# Contribution of Blood Vessel Activation, Remodeling and Barrier Function to Inflammatory Bowel Diseases

**DOI:** 10.3390/ijms24065517

**Published:** 2023-03-14

**Authors:** Nathalie Britzen-Laurent, Carl Weidinger, Michael Stürzl

**Affiliations:** 1Division of Surgical Research, Department of Surgery, Translational Research Center, Universitätsklinikum Erlangen, Friedrich-Alexander-Universität Erlangen-Nürnberg (FAU), 91054 Erlangen, Germany; 2Comprehensive Cancer Center Erlangen-EMN (CCC ER-EMN), 91054 Erlangen, Germany; 3Department of Gastroenterology, Infectious Diseases and Rheumatology, Charité-Universitätsmedizin Berlin, Campus Benjamin Franklin, 12203 Berlin, Germany; 4Division of Molecular and Experimental Surgery, Translational Research Center, Universitätsklinikum Erlangen, Friedrich-Alexander-Universität Erlangen-Nürnberg (FAU), 91054 Erlangen, Germany

**Keywords:** inflammatory bowel diseases (IBDs), vasculature, angiogenesis, gut vascular barrier, vessel permeability

## Abstract

Inflammatory bowel diseases (IBDs) consist of a group of chronic inflammatory disorders with a complex etiology, which represent a clinical challenge due to their often therapy-refractory nature. In IBD, inflammation of the intestinal mucosa is characterized by strong and sustained leukocyte infiltration, resulting in the loss of epithelial barrier function and subsequent tissue destruction. This is accompanied by the activation and the massive remodeling of mucosal micro-vessels. The role of the gut vasculature in the induction and perpetuation of mucosal inflammation is receiving increasing recognition. While the vascular barrier is considered to offer protection against bacterial translocation and sepsis after the breakdown of the epithelial barrier, endothelium activation and angiogenesis are thought to promote inflammation. The present review examines the respective pathological contributions of the different phenotypical changes observed in the microvascular endothelium during IBD, and provides an overview of potential vessel-specific targeted therapy options for the treatment of IBD.

## 1. Introduction

Inflammatory bowel diseases (IBD) are a group of intestinal chronic inflammatory disorders characterized by cyclic flares of destructive inflammation that comprise two major forms, Crohn’s disease (CD) and ulcerative colitis (UC) [1]. Although inflammation in UC is restricted to the colon and only extends to the mucosal layers, transmural inflammation can be observed in CD, which can manifest at any site of the gut. The etiology of IBD is thought to be multifactorial and to involve the patient’s genetics and immune response, the intestinal microbiome, and environmental factors [2]. IBD is considered to result from an inappropriate immune response to the intestinal microflora and environmental triggers in genetically susceptible individuals [1]. The resulting inflammation induces tissue damage and, notably, a disruption of the epithelial barrier, leading to the perturbation of the intestinal microenvironment and the relationship between the mucosal surface and the commensal microbiota. This disequilibrium not only affects the maintenance, function, and repair of the epithelial barrier, it also results in the perturbation of the commensal microbiota composition, leading to dysbiosis [3,4].

The intestinal microvasculature lies in close vicinity to the epithelial layer and represents a second barrier to the penetration and dissemination of commensals and microbial products. In IBD, the microvascular endothelium is strongly affected by inflammation [5]. Endothelial cells (ECs) of the gut microvasculature are activated to allow leukocyte recruitment and infiltration, whereas the vascular barrier function is compromised and local bursts of angiogenesis are observed. Despite its well-defined function in inflammation, the contribution of the endothelium to the development and maintenance of IBD has been rather overlooked, in particular as a potential therapeutic target. In the present review, we summarize the respective contributions of blood vessel activation, remodeling, and barrier function to the pathogenesis of IBD, and discuss the status and perspectives of vessel-directed therapies.

## 2. The Intestinal Vasculature in Homeostasis

The intestine is vascularized by arterioles from the submucosa, which divide into capillary networks in the mucosa and muscle layers, with the mucosal layers receiving 80% of the total blood flow [6]. The anatomy of the mucosal vasculature differs between the small and large intestine (colon) due to their different tissue architectures. In the small intestine, the epithelium builds villi and crypts. Each villus contains a single arteriole going to the tip, forming a tuft-like network of capillaries which are located directly under the epithelial monolayer, and the blood is collected into a single central venule. The crypts are infused with a capillary network which drains into the venule as well. In contrast to the small intestine, the colon epithelium does not have villi. The arterioles and their capillary branches are arranged along the colonic crypts, developing into a capillary honeycomb-like network around the crypts that is in very close proximity to the epithelial layer (1 μm) [6,7]. Intestinal post-capillary veins are devoid of smooth-muscle cells and represent the most reactive segment of the microvasculature [8]. The intestinal microcirculation regulates oxygen and nutrient exchange, tissue fluid homeostasis, and leucocyte abundance [9,10,11].

Under physiological conditions, the single layer of ECs lining the vessel lumen provides an anti-adhesive and selectively permeable exchange barrier. In the past few years, the general opinion about the role of the intestinal microvasculature has evolved, and it is now viewed as an integral component of the intestinal barrier [12]. The intestinal barrier is tightly regulated to allow the absorption of essential nutrients, electrolytes, and water from the intestinal lumen into the circulation, while preventing the entry of microbiota through different layers of protection. The first barrier is formed by a tight epithelial monolayer covered by a thick layer of mucus produced by specialized enterocytes, the goblet cells. In addition, another population, the Paneth cells, secretes antimicrobial peptides. This intestinal epithelial barrier prevents the penetration of microbes or microbe-derived molecules into the tissue. A second barrier, the gut vascular barrier (GVB), has been identified in humans and mice, and provides a second layer of protection, which blocks microbial dissemination into the systemic circulation (Figure 1) [12,13,14].

In contrast to the blood–brain barrier (BBB), which has a size exclusion threshold of 500 Da, the GVB is permeable to molecules as large as 4 kDa, allowing the passage of nutrients and antigens for tolerance induction [14]. Spadoni et al. demonstrated that endothelial cells in the intestine are closely associated with pericytes and enteric glial cells to form what they termed gut–vascular units (Figure 1) [13]. Enteric glial cells seem to be crucial for the development and maintenance of the GVB, as it has been shown that transgenic mice lacking enteric glial cells feature increased epithelial permeability and microvascular disturbances, resulting in the uncontrolled spread of bacteria into the blood circulation and the subsequent death of the affected animals (Table 1) [15,16]. Interestingly, the interplay of the GVB with the commensal microbiota was found to increase angiogenesis, endothelial coverage, and the formation of the enteric glial cell network in the lamina propria using human ECs and murine models [17,18,19,20], suggesting that the microbiota supports the formation and maintenance of the GVB.

Within the gut–vascular units, endothelial cells form tight cell–cell contacts, which are enhanced through interaction with enteric glial cells and pericytes. Adherens junctions (AJs) and tight junctions (TJs) found in human and mouse intestinal microvascular ECs regulate paracellular trafficking of molecules and leukocytes and express different classes of transporters (such as ATP-binding cassette transporters and sugar transporters) [12]. TJs control permeability for ions and small molecules (<800 Da), whereas AJs are primarily responsible for the maintenance of vascular barrier function and control its permeability for molecules of high molecular weight [21]. In intestinal ECs, AJs are composed of vascular endothelial cadherin (VE-cadherin), α- and β-catenin, and p120, all of which are expressed homogenously throughout the intestines and the vascular beds. The building of VE-cadherin adhesions is considered to be the primary event during vascular development [13,21]. It precedes TJ formation and is required for TJ maintenance. Disruption of AJs leads to disassembly of TJs [13,21]. Although numerous TJs can be found in small arterioles, the level of TJs is reduced in capillaries and post-capillary veins. As a result, AJs are predominantly found in capillaries and post-capillary venules [21]. In intestinal ECs, TJs are formed by occludin, zonula occludens-1 (ZO-1), cingulin, junctional adhesion molecule-A (JAM-A), and claudins [13,22]. The expression of endothelial claudins has been extensively studied in the mouse intestine and varies between gut areas and cell types. While claudin-1 is expressed at similar levels in ECs throughout the intestines, the channel-forming claudins-7, -12, and -15 are exclusively expressed in the colon [22]. Outside of the BBB and the blood–retinal barrier, where a high expression of claudin-5 prevents the passage of small molecules, claudin-5 exhibits diminishing expression along the arteriovenous axis [23]. In the gut, claudin-5 expression is restricted to lymphatic ECs, high endothelial venules (HEVs), and certain capillary ECs [13,24]. These different patterns of expression are thought to be responsible for variations in permeability and to reflect the site-specific physiological function of the GVB. In humans, claudins working as channels show a higher expression in the colon, where they regulate solute paracellular transport, compared to the small intestine or the BBB [25,26].

Kalucka et al. performed a single-cell transcriptome analysis of murine ECs across 11 tissues, revealing heterogeneity between tissues and vascular beds [24]. Overall, they found that colon and small-intestine ECs are characterized by the high expression of genes involved in vascular barrier integrity and maintenance. In addition, two specific EC-fractions were found in the intestine. First, a subset of capillary ECs was described, which display an elevated expression of genes involved in the uptake and metabolism of glycerol and fatty acids. Cells in this cluster were notably characterized by a high expression of aquaglyceroporin 7 (*Aqp7*), a pore-forming transmembrane protein involved in glycerol transport across cell membranes, and were therefore termed Aqp7+ capillary ECs. Second, a subset of intestinal venous ECs showing enriched expression of HEV markers (*Madcam1*, *Lrg1*, *Ackr1*) was identified. HEVs represent a specialized subtype of post-capillary venules mediating the recruiting and trafficking of lymphocytes from blood to lymph nodes and secondary lymphoid organs, which take their name from the cuboidal appearance of their endothelial cells (Table 1).

The investigation of isolated human intestinal mucosa-derived endothelial cells (HIMECs) has revealed unique functional features when compared to human umbilical veins (HUVECs) (Table 1). For instance, exposure to lipopolysaccharide (LPS) induces a transient increase in the presence of adhesion molecules in HIMECs, compared to a long-lasting increase in the presence of adhesion molecules in HUVECs [27]. This reflects the relative tolerance of intestinal microvascular cells to bacterial products from the gut microbiota. In addition, HIMECs produce different cytokines, including IL-3 and IL-6, as HUVECs upon activation with inflammatory cytokines [28]. Furthermore, HIMECs constitutively express the (otherwise) inducible nitric oxide synthase (iNOS) in addition to endothelial NOS (eNOS) [29]. Under physiological conditions, endothelial-derived NO maintains the anti-adhesive state of the endothelium by limiting leukocyte and platelet adhesion, and regulates vasodilatation [29,30].

Overall, in the state of homeostasis, intestinal ECs are involved in barrier function maintenance, nutrient uptake, and immune tolerance.

## 3. The Intestinal Vasculature in IBD

The intestinal microvasculature plays a crucial role during inflammation by regulating tissue recruitment of inflammatory cells and wound healing. However, uncontrolled inflammation induces a sustained EC activation (Figure 1), causing an increase in leakiness (edema), adhesiveness (leukocyte recruitment), pro-coagulant activity (thrombus), and angiogenesis (immature vessels) [5,31]. As a result, inflammation is enhanced, ultimately leading to sustained tissue and vessel damage.

The histopathological analysis of inflamed human and murine intestinal tissues has revealed massive changes in the blood microvasculature, including vasodilatation, vasocongestion, edema, flares of angiogenesis, microvascular occlusions, and abnormal vessel architecture characterized by tortuous vessels of varying diameter (Figure 2) [32,33,34]. These profound alterations have been considered to be an early event since they precede the development of mucosal ulceration, and to significantly enhance inflammation in IBD [32]. Hence, the intestinal microvasculature can be seen both as a regulating factor and as a target of inflammation. In the following, we examine the respective contributions of intestinal vascular changes to IBD pathogenesis (Figure 2).

### 3.1. Endothelial Cell Activation and Leukocyte Recruitment

The recruitment of circulating leukocytes into tissues is an early and central event during inflammation. It starts with the activation of the microvascular endothelium by inflammatory mediators, including cytokines such as IFN-γ, IL-1β, and TNF-α (Table 2). The activated endothelium regulates the leukocyte extravasation cascade in a tightly coordinated sequence, including tethering and rolling, activation, adhesion, spreading, and the transmigration of leukocytes [35]. Activated ECs are characterized by an elevation in the level of cell adhesion molecules (CAMs), the production of chemokines, and the expression of costimulatory molecules, which further amplify the recruitment and activation of leukocytes (Figure 1). Leukocyte hyper-adhesion has been observed in the intestinal ECs of patients with IBD. For instance, Binion et al. described a significant increase in leukocyte binding in HIMECs isolated from inflamed regions of IBD patients compared to HIMECs that were obtained from non-inflamed intestinal sites or from the guts of control subjects [36].

At the molecular level, the MAPK pathway has been shown to play an important role in the upregulation of CAMs and the production of chemokines by activated HIMECs, as well as lymphocyte extravasation [37]. In the inflamed mucosa of IBD patients, increased levels of phosphorylated MAPK have been detected in the microvasculature [37].

Among the various CAMs expressed in activated ECs, P- and E-selectins are glycoproteins involved in the rolling and recruitment of leukocytes. Although P-selectin is constitutively available as a pool that can be mobilized upon activation, E-selectin expression is induced in response to inflammatory stimuli. A notable increase in P-selectin levels has been observed in the colons of UC patients compared to controls, whereas serum levels of the decoy soluble P-selectin were decreased [37,38,39].

An important class of adhesion molecules expressed in ECs is the immunoglobin CAM superfamily, which includes intracellular cell adhesion molecule 1 (ICAM-1), vascular cell adhesion molecule 1 (VCAM-1), platelet endothelial cell adhesion molecule (PECAM-1, also known as CD31), as well as the gut-specific mucosal addressin cell adhesion molecule 1 (MadCAM-1). Upon EC activation, ICAM-1 is recruited from the EC junctions to the apical surface [27]. The microvascular expression of ICAM-1 is increased in IBD patients, and ICAM-1 has been shown to be crucial to T-cell recruitment in the T-cell transfer murine colitis model [40]. ICAM-1 is constitutively expressed in HIMECs and can be upregulated by inflammatory cytokines and vascular endothelial growth factor-A (VEGF-A) [27,41,42]. VCAM-1, which mediates adhesion to lymphocytes expressing integrin α4β1 or α4β7, is also inducible in HIMECs, notably by VEGF-A. The expression of VCAM-1 in the mucosal vasculature is increased in patients with IBD and murine colitis models, where its expression correlates with disease severity [43,44]. PECAM-1/CD31 is also inducible by inflammatory cytokines and is involved in leukocyte rolling and firm adhesion during IBD [45]. A large amount of attention has been paid to MadCAM-1, a gut-specific homing molecule mediating the recruitment of T and B cells expressing integrin α4β7 [46]. High MadCAM-1 expression has been observed in the inflamed intestinal endothelium during IBD [36]. In HIMECs, MadCAM-1 expression can be induced by inflammatory cytokines (IFN-γ, IL-1β, TNF-α) [47,48]. Interestingly, the expression of MadCAM-1 in HIMECs is inversely correlated to cellular density, suggesting that high MadCAM-1 expression might be a marker of proliferating vessels [47,48]. Recently, MadCAM-1 has been found to be critical for the recruitment of antibody-producing B cells into the intestinal mucosa in the IL-10-knockout colitis model [49].

### 3.2. Pathological Angiogenesis in IBD

Angiogenesis is a hallmark of chronic inflammation [50]. In human IBD and in several murine models of colitis, the microvascular density is increased and directly correlates with disease severity (Figure 1 and Figure 2) [51,52,53]. Angiogenic ECs exhibit increased proliferation and migration, as well as a unique cell-surface molecular pattern. For instance, the integrins ανβ3 and ανβ5 are specifically expressed at the surface of ECs from newly formed vessels, and an increased expression of ανβ3 has been observed in the inflamed mucosa of IBD patients [51]. The blockade of ανβ3 was found to reduce disease activity in the IL-10 knockout colitis model, suggesting that angiogenesis contributes to IBD pathogenesis [54]. The angiogenic expansion of the vascular bed is assumed to physically increase blood supply through the increased endothelial surface, and therefore to enhance the leukocyte supply to the tissue. However, the newly formed vessels in the context of chronic inflammation show an immature phenotype in mouse and human tissues [34,55,56]. They are leaky, have less or no coverage by pericytes, and are hypoperfused and often hyperthrombic [32,33,34]. Stenoses are also frequently observed [32]. Hence, pathological angiogenesis appears to contribute to the intense vascular remodeling observed in IBD.

The fact that mucosal extracts of IBD patients could induce dose-dependent HIMEC migration in vitro supported the presence of a pro-angiogenic microenvironment in the inflamed gut mucosa [57]. Two main mechanisms of angiogenesis have been proposed to occur during IBD, namely, extension from existing vessels (sprouting) and vessel splitting (intussusception). Angiogenesis through the recruitment of endothelial progenitor cells seems less likely to occur during colitis since the number of bone-marrow-derived endothelial progenitor cells is reduced in UC [58,59,60].

The induction of sprouting angiogenesis during IBD has been partly attributed to inflammation-related hypoxia, with hypoxia-inducible factor-1 and -2 transcriptionally activating the expression of vascular endothelial growth factor A (VEGF-A), the major angiogenic growth factor (AGF) [61]. In agreement with this hypothesis, the human inflamed intestinal epithelium was found to represent an important source of VEGF-A, which is also produced by leukocytes [62,63,64,65]. Furthermore, increased expression of VEGF and other AGFs including basic fibroblast growth factor (bFGF) and platelet-derived growth factor (PDGF) has been detected in mucosal extracts and in the serum of IBD patients as compared to controls (Table 3) [51,53,66,67,68,69,70], although this increase was more evident for UC than for CD, especially for VEGF [61,71,72,73]. However, experimental colitis models have provided conflicting results regarding the contribution of sprouting angiogenesis to disease activity. The inhibition of angiogenesis via the neutralization of VEGF-A improved the course of intestinal inflammation in mice [34,64,74,75] and modestly in rats [76]. In contrast, the knock-out of placental growth factor, a VEGF homolog, also caused decreased angiogenesis, but lead to an aggravation of colonic injury in the mouse dextran sodium sulfate (DSS)-induced colitis model [77]. These results are reflected in the responses of cancer patients treated with anti-angiogenic treatment. Rare adverse effects of bevacizumab, a humanized monoclonal antibody against VEGF, include intestinal perforation, gastrointestinal bleeding, and ulcerative colitis [78,79,80]. However, anti-VEGF therapy is well tolerated by most patients with quiescent and moderately active IBD [81]. The frequency of inflammatory side effects is higher when the PDFG pathway is involved. For instance, an exaggeration of UC has been observed during treatment with the angiogenic inhibitors sunitinib and sorafenib, which target both the VEGF and PDGF pathways [82,83,84].

Several reasons might explain the contradictory effects of angiogenesis in IBD. Firstly, while angiogenic and inflammatory vessels can synchronously co-exist in the inflamed mucosa during mouse colitis [32], the two phenotypes are mutually exclusive in a single EC. In particular, inflammatory cytokines (ICs) such as IFN-γ, TNF-α, and IL1-β can inhibit AGF-induced proliferation and the migration of human ECs in vitro (Table 2) [85,86,87]. In addition to ICs, several anti-angiogenic factors are upregulated during intestinal inflammation, including the chemokine CXCL-10, thrombospondin, angiostatin—a cleaved fragment of plasminogen, and endostatin—a cleaved fragment of collagen XVIII [63,88,89,90,91,92]. Hence, the balance between angiogenic and inflammatory-associated angiostatic stimuli might explain the relative contribution of angiogenesis to IBD pathogenesis.

Secondly, there is a strong interplay between angiogenesis, inflammatory vessel activation, and barrier function during inflammation. For instance, human and murine VEGF-A stimulates angiogenesis and increases vessel permeability, while reducing vessel coverage [41,42,93]. In murine colitis models, the permeability marker CD146/MUC18 was shown to induce angiogenesis, lymphangiogenesis, and leukocyte recruitment [94,95,96]. In a similar manner, mice that are deficient in CD40 or CD40L display both a decrease in leukocyte and platelet recruitment and impaired angiogenesis in the gut [97,98,99].

Thirdly, the importance of intussusceptive angiogenesis during IBD might have been underestimated. The vessel splitting that is characteristic of intussusceptive angiogenesis occurs through intraluminal endothelial cell rearrangements rather than endothelial cell proliferation [100]. These are triggered by increased blow flow and are regulated by the nitric oxide (NO), endoglin, and ephrinB2/EphB4 signaling pathways [101,102,103,104,105]. Intussusceptive angiogenesis has been observed in murine colitis [100,106,107], where it was induced through mechanical forces and changes in the intraluminal blood flow, and was regulated by MT1-MMP [100,106]. EC-specific knockout of MT1-MMP ameliorated dextran sodium sulfate (DSS)-induced colitis in mice [107]. The predominance of intussusceptive angiogenesis compared to sprouting angiogenesis during colitis might explain the relative success of classical anti-angiogenic approaches in animal models of IBD.

Finally, angiogenesis not only sustains inflammation, it also plays an essential role during mucosal healing. For instance, VEGF-A is involved in UC healing and angiogenesis via the recruitment of cells expressing vascular endothelial growth factor receptor 1 (VEGFR1), including monocytes, Tregs, and bone-marrow derived stem cells, to ulcerated tissues [108]. Furthermore, the expression of the Wnt pathway member adenomatous polyposis coli (APC) in murine intestinal ECs has been shown to mediate mucosal repair following colonic inflammation through angiogenesis [109]. Hence, treatment with angiogenesis inhibitors might result in wound healing complications. Despite the activation of angiogenic signals, impaired mucosal healing is observed and represents a major issue in IBD, notably in UC. Pathological angiogenesis or an increase in anti-angiogenic signals such as endostatin and angiostatin might explain why mucosal lesions are slow to repair in UC [91]. In addition, the decrease in the number of bone-marrow-derived endothelial progenitor cells (BMD-EPC) observed in UC, whether it is due to a decreased release from the bone marrow and/or impaired homing in colonic lesions, participates in mucosal healing impairment [59,60]. In addition, there is crosstalk between VEGF and transforming growth factor-beta (TGF-β), which is essential for wound healing, tissue repair, and the resolution of inflammation. A dysregulation of the TGF-β pathway, as seen in case of the impairment of endoglin, the endothelial-specific co-receptor for TGF-β, caused the levels of VEGF to spike in the acute DSS colitis model [110,111]. This resulted in enhanced and chronic intestinal inflammation, characterized by higher angiogenesis and MAdCAM-1 vascular expression [110,111]. Hence, both the therapeutic inhibition of angiogenesis and the presence of exacerbated or pathological angiogenesis seem to impair wound healing in IBD and experimental colitis models.

Another example of the interplay between vascular remodeling and inflammation in IBD is given by HEVs (Table 1). Subsequently to the ectopic formation of tertiary lymphoid organs observed in the inflamed gut mucosa, the density of HEVs has been found to increase in the mucosa of IBD patients [112,113]. However, HEVs not only regulate the lymph drainage of antigen-presenting cells, they also regulate the homing of T-cells. Extrafollicular HEV formation has been observed in the intestinal mucosa of IBD patients during active inflammation, where it correlates with T-cell infiltration and disease activity [113,114]. Human intestinal HEVs express high levels of MadCAM-1, and in UC, the O-glycosylation of MadCAM1 is increased, which induces a higher binding to L-selectin, expressed on leukocytes [112,115]. Taken together, pathological angiogenesis in IBD seems to both potentiate inflammation and impair mucosal healing.

### 3.3. The Gut–Vascular Barrier in IBD

During IBD, the epithelial barrier is compromised, as shown by an increased permeability and mucosal damage such as erosions or ulcers [116,117]. In consequence, the passage of antigens, bacteria, and bacterial products into the submucosa is increased [117]. This further enhances the local inflammation, resulting in the release of large amounts of inflammatory mediators from—but not restricted to—immune cells, which can in turn affect the gut–vascular barrier [118].

IBD is associated with increased gut vascular permeability (Figure 1), which is indicative of a loss of intestinal vascular barrier function, and results in edema and tissue damage [33,34,69,119,120,121]. For instance, human CD146 (MUC18), a cell junction molecule constitutively expressed in ECs, and its soluble form (sCD146), which is considered a marker for vascular permeability, are upregulated in intestinal ECs and serum of IBD patients, respectively [96,122]. Endothelial damage and increased colonic vascular permeability have been observed early during the development of experimental ulcerative colitis in rats and mice [123]. Post-capillary venules, the most reactive part of the vascular tree, were also shown to be the major site of vascular leakage [124]. The increased vascular permeability of inflamed post-capillary venules has been attributed to endothelial cell-cell contact disruption, to the contraction of activated ECs, to EC death and detachment, and/or to plasma protein extravasation at the site of leukocyte transendothelial migration [125,126]. A variety of inflammation mediators found in IBD can induce vascular permeability, including histamine, serotonin, substance P, bradykinin, and ICs, notably IFN-γ and TNF-α (Table 2) [121]. In addition, the reduction of anti-inflammatory cytokines such as IL-10 can further amplify intestinal vascular permeability induced by IFN-γ in experimental colitis [121]. Vascular permeability is also increased during angiogenesis, notably through a direct effect of VEGF [127,128,129]. Intussusceptive angiogenesis has also been associated with an enhanced permeability, likely because it induces holes in the vascular layer. In particular, endothelial cell-specific MT1-MMP knockout mice, which are characterized by a lower intussusceptive angiogenesis, also display a reduced vessel permeability during DSS-induced colitis [107].

In a study comparing the respective effects of IFN-γ and VEGF on disease development, both endothelial-specific knockout of the IFN-γ receptor (IFNγR) and VEGF blockade inhibited DSS-induced colitis in mice [34]. In agreement with the angiostatic properties of IFN-γ and the pro-angiogenic effect of VEGF, angiogenesis was increased in the case of IFNγR knockout, whereas it was decreased after VEGF blockade as compared to controls. Both approaches, however, led to a strong decrease in vascular permeability [34,74]. These results suggested that the induction of vascular permeability by VEGF might contribute more to colitis pathogenesis than the mere induction of angiogenesis. Several reports further support the promoting role of GVB disruption in IBD. Vascular permeability is associated with disease activity in UC and CD [34]. The inhibition of vascular permeability with the RTK inhibitor imatinib restores VE-cadherin junctions, increases vascular coverage, and inhibits DSS-induced colitis [34]. Fibrinogen, which is upregulated in UC and mouse colitis, was shown to promote DSS-induced colitis by enhancing vascular permeability [130]. Moreover, transient receptor potential vanilloid 4 (TRPV4) channels were found to enhance DSS-induced colonic inflammation in mice through an increase in vascular permeability [131]. Taken together, these studies have established the pathogenic contribution of vascular barrier breakdown to IBD.

#### 3.3.1. VE-Cadherin and Vascular Barrier Regulation in IBD

VE-cadherin, the major component of endothelial adherens junctions and the master regulator of vascular barrier function, is regarded as the primary target during inflammation-induced vascular permeability [132,133,134]. In IBD patients, membrane VE-cadherin expression is significantly reduced in blood vessels found in inflamed areas compared to uninvolved intestinal tissues [34]. Several mechanisms of VE-cadherin junction disruption have been described. Inflammation-induced proteases such as matrix metalloproteinases (MMPs) or elastase, which are produced by leukocytes, smooth muscle cells, and ECs, can promote vascular permeability through the degradation of VE-cadherin AJs and TJs [5,34,135,136,137,138]. In IBD, vascular smooth muscle cells and pericytes express MMP-1 and MMP-9 [137], and MMP-9 serum levels are increased [139,140,141]. This is supported by the fact that MMP9 deficiency was found to attenuate intestinal injury in animal colitis models [142,143,144]. In addition, MMP-catalyzed VE-cadherin cleavage results in the generation of soluble VE-cadherin, which itself can further destabilize the vascular barrier by impairing the binding of VE-cadherin molecules, as shown in human rheumatoid arthritis, systemic inflammation, and sepsis [145,146,147].

The vascular hyperpermeability induced by VEGF and inflammatory cytokines also involves the direct disassembly of VE-cadherin junctions. Binding of VEGF to VEGFR2 at the surface of human ECs activates the Src kinase, resulting in VE-cadherin phosphorylation and internalization, which occurs via clathrin-dependent endocytosis and is mediated by neuropilin and Rac [127,129,147,148,149,150]. In a similar manner, TNF-α was shown to increase tyrosine phosphorylation of VE-cadherin and to open the paracellular pathway in the human lung endothelium through the activation of the Fyn kinase [138]. The effect of IFN-γ on the disruption of the VE-cadherin junction was found to be even stronger and longer lasting than the effect of VEGF in human ECs and mouse intestinal endothelial cells (MIECs) [33,34,151]. However, the molecular mechanism by which IFN-γ dismantles VE-cadherin junctions remains to be elucidated.

VE-cadherin disruption can also result from EC contraction, since the VE-cadherin complex and the actin cytoskeleton are functionally connected [134,152,153]. For instance, treatment of ECs with TNF-α was found to induce an almost immediate rise in mechanical substrate traction force and internal monolayer tension [154]. VEGF also induces actin reorganization and the migration of endothelial cells via the serine/threonine kinase Akt [155]. A similar disruption of actin has been observed in human ECs following exposure to IFN-γ, and might explain, at least partially, the IFN-γ-induced destabilization of VE-cadherin [156]. Such a mechanism was also observed during experimental colitis, where fibrinogen was shown to induce vascular permeability through activation of AKT and depolymerization of actin microfilaments [130].

Downregulation of AJ and TJ adhesion molecule expression has also been observed upon exposure to ICs and VEGF. For instance, VE-cadherin expression decreases in MIECs after treatment with IFN-γ [34]. Using an in-vitro intestinal endothelial barrier model composed of rat intestinal microvascular endothelial cells, Liu et al. found that TNF-α decreases the expression of TJ proteins, including ZO-1, occludin, and claudin-1, while increasing the expression of pore-forming claudin-2 [157]. Nevertheless, several reports showed no difference in the TJ-associated protein expression of zonula occludens-1 (ZO-1) in MIECs or intestinal endothelial cells during DSS-induced colitis [34,158]. In this model, ZO-1 was notably only reduced in epithelial cells but not in ECs [158], supporting the predominant role of VE-cadherin junctions in the regulation of the GVB.

#### 3.3.2. Vessel Coverage and Permeability

During intestinal inflammation, vessel coverage with adventitial support cells, the so-called mural cells (pericytes and smooth muscle cells), is reduced in mouse models of colitis [159,160]. Mural cell recruitment is regulated by the PDGF-B-PDGFR-β pathway. PDGF-B is secreted by sprouting ECs and signals through PDGFR-β at the surface of mural cells [159]. This induces the proliferation and migration of mural cells. PDGF-B/PDGFR-β knockout models in mice showed reduced mural cell coverage and increased vessel permeability [159,161,162]. In the DSS colitis model, vessel coverage has been found to promote the stabilization of the vascular barrier, decreasing vessel permeability and inflammation [163,164]. Endothelial IFN-γ receptor knockout leads to increased vessel coverage during DSS-induced colitis in mice [33,34]. This could be attributed to the fact that IFN-γ inhibits PDGF-like protein release and decreases the PDGF-B chain mRNA level in HUVECs [165]. Similarly, excess VEGF-A disrupts pericyte recruitment and VEGF blockade was found to increase vessel coverage in the DSS-colitis model [34,166].

### 3.4. Microvascular Dysfunction in IBD

In IBD, the mucosal vasculature displays pathological traits, including tortuous structures, edema, arteriolar dilatation, hypercoagulation, and vascular damage (Figure 2) [32]. These vascular defects are multifactorial. For instance, chronic high levels of angiogenic factors and inflammatory cytokines can alter microvascular structure and function [167]. More precisely, structural changes in vessels have been shown to result from pathological angiogenesis and inflammation-driven GVB dysfunction, including the loss of mural cell coverage and an immature phenotype [167]. Another example is edema formation, which results both from increased afferent blood flow (hyperemia) due to arteriolar dilatation and from hyperpermeability. The imbalances in vascular function observed during IBD lead to complex and sometimes contradictory effects. In IBD patients, for instance, hyperemia has been observed in submucosal arterioles enlarged by increased afferent blood flow, whereas decreased perfusion was found in the intestinal mucosa [168].

#### 3.4.1. The Role of Nitric Oxide in Vascular Dysfunction during IBD

Among the numerous mediators involved in microvascular function, nitric oxide (NO) has attracted a large amount of interest due to its broad range of functions. Under physiological conditions, NO counteracts leukocyte and platelet adhesion to ECs, regulates vasodilatation and endothelial permeability, and acts as radical scavenger [30]. As mentioned above, HIMECs express both the endogenous endothelial and the inducible form of NO-synthase (eNOS and iNOS), indicating the high tolerance of the gut microvasculature towards inflammatory activation [29]. The resulting high NO levels can, for instance, inhibit the expression of endothelial CAMs and MMPs induced by ICs (Table 3). During IBD, the production of NO by ECs is reduced due to the loss of iNOS and eNOS in HIMECs, and leukocyte adhesion is increased [169,170]. Furthermore, an upregulation of arginase expression and activity has been observed in inflamed HIMECs. Arginase competes with NOS for L-arginine, and therefore can limit NO production due to reduced substrate availability [170]. The decreased production of NO in ECs during IBD results in a loss of NO-mediated vasodilatation and an increase in ROS production in the microvessels of affected intestinal areas [171,172]. In parallel, NO also plays a role in VEGF-driven angiogenesis, as well as in intussusceptive angiogenesis [107], hence contributing both to the perpetuation of inflammation and to wound healing. Finally, NO has been shown to regulate endothelial barrier function in human, murine, and bovine ECs, notably by promoting VEGF-induced permeability through targeting of the VE-cadherin/β-catenin and Rho pathways [173,174]. However, NO can also protect ECs from hypoxia-induced barrier dysfunction [175]. In line with these contradictory results, the deficiency of eNOS and iNOS has been associated with either a better or a more severe course of disease in mouse models of colitis [176,177,178,179]. These differences might be attributed to the different roles played by NO in different cell compartments. The expression of eNOS by intestinal endothelial cells has been shown to specifically maintain mucosal integrity and prevent bacterial translocation in an TNBS-colitis model in mice [179]. Overall, the impairment of NO production during IBD increases the inflammatory activation of ECs and impairs vasodilatation, which might ultimately lead to pathological vasoconstriction, reduced mucosal perfusion, impaired wound healing, and the maintenance of chronic inflammation.

#### 3.4.2. Coagulation

In mucosal tissues from patients with IBD or murine colitis models, the presence of thrombi due to increased platelet activation and binding to the EC surface has been observed [57]. This, in turn, can result in ischemic inflammation in the intestinal microvasculature, further enhancing tissue damage. In the DSS-colitis model and in IBD patient samples, accumulation of platelets in mucosal venules is linked to an increased leukocyte binding and disease activity [57,180,181]. The increase in thrombus formation associated with IBD has been attributed to several factors (Table 3). At the endothelial level, the activation of ECs through ICs and the decrease in NO levels upregulate the surface expression of adhesion molecules. Activated platelets found in the general circulation and in the intestinal mucosa of IBD patients can activate ECs, notably through the expression of P-selectin or CD40L, as well as the release of the soluble form of CD40L (sCD40L) [182,183]. In turn, P-selectin and CD40L/sCD40L activate HIMECs, resulting in increased adhesion molecule expression, the secretion of cytokines such as IL-8, and increased binding to leukocytes, as well as to platelets themselves [97,183]. Increased coagulation has also been attributed to a reduced expression of thrombomodulin and protein C receptor (PCR) in the mucosal microvasculature of IBD patients and in colitic mice [184,185,186,187]. Protein C activation is also impaired in HIMECs under inflammatory conditions. This dampening of the PC system is correlated with increased adhesiveness of ECs, thereby promoting leukocyte recruitment and inflammation.

### 3.5. Regulatory Role of the Vasculature during Mucosal Inflammation

#### 3.5.1. Vasculature and Innate Immunity

Following the rupture of the gut–epithelial barrier, mucosal microvascular ECs are exposed to bacteria and bacterial products. ECs can then launch an innate immune reaction through the engagement of toll-like receptors (TLRs) via pathogen-associated molecular patterns (PAMPs). Gut microvascular ECs exhibit a particular TLR response pattern compared to ECs from other origins (Table 3). For instance, tolerance to lipopolysaccharide (LPS), an activator of TLR4, has been observed in HIMECs but not in HUVECs after repeated exposure [188]. TLR3 and TLR5 are expressed at the surfaces of intestinal ECs. TLR3 is involved in the anti-viral response and is constitutively expressed in HIMECs, where it can be further upregulated by IFN-γ [189]. TLR5, a receptor for flagellin, was shown to play an important role in the EC innate immune response [190,191]. Activation of TLR3 and TLR5 in HIMECs induces the upregulation of inflammatory mediators and ICAM-1, leading to leukocyte recruitment [189,192]. TLRs expressed by intestinal ECs serve as a second barrier in the case of epithelial barrier breakdown. In particular, flagellin has been described as a dominant antigen in CD [193,194,195]. In agreement with this protective function of the GVB, TLR3 and TLR5 expression on ECs is protective against colitis in mice [196,197]. Taken together, the propensity of HIMECs to develop endotoxin tolerance might prevent an excessive immune reaction from occurring in the case of luminal bacterial penetration into the mucosa, while the response to viruses and flagellin remains intact, protecting against systemic propagation.

#### 3.5.2. Paracrine Effects of the Inflamed Vasculature

ECs are highly reactive and can themselves express and secrete inflammatory mediators upon activation [192,198,199]. Those factors can either further promote or dampen inflammation through local or systemic effects (Table 3). For example, CX3CL1 (fractalkine), a chemokine that is upregulated in ECs via the MAPK pathway upon stimulation by TNF-α, IL-1, LPS, and IFN-γ, is highly upregulated in the mucosal endothelia of IBD patients [200,201]. CX3CL1 released from ECs stimulates the adhesion and transmigration of leukocytes expressing the CX3CR1 receptor. Higher levels of circulating and infiltrating CX3CL1^+^ T cells have been observed during IBD [202]. Under inflammatory conditions, HIMECs are also able to express monocyte chemotactic protein 1 (MCP-1) [199]. In addition, intestinal vessels might exert angiocrine activity on epithelial cells during IBD through the release of TIMP1 and CXCL10, the latter being able to increase crypt survival [199].

Vascular ECs also secrete the C-C chemokine ligands 5 (CCL5) and 25 (CCL25) during inflammation [5]. CCL25 leads to the recruitment of immune cells expressing CCR9 [203]. Both CCR9 and CCL25 are upregulated during DSS-induced colitis, in which the CCL25-CCR9 axis exerts a protective anti-colitic effect in the intestinal mucosa by balancing different dendritic cell subsets [203]. Furthermore, during experimental colitis, NO of endothelial origin protects the intestinal mucosa of mice against inflammation by increasing the number of goblet cells and mucin production, thereby preventing luminal bacteria translocation [179].

### 3.6. Role of the Mesenteric Lymphatic Vasculature in IBD

Intestinal lymphatic vessels are involved in the removal of excess interstitial fluids, immune regulation, and the absorption of fatty acids by lacteal lymphatic endothelial cells [204]. During inflammation, lymphatic endothelial cells participate in the regulation of the adaptive inflammatory response and promote the resolution of inflammation through the clearance of immune cells and mediators [204]. Lymphangiogenesis is frequently observed during inflammation [205]. An increase in lymphatic vessel density has been observed in UC and CD throughout the mucosa, including in non-inflamed areas [206,207,208]. This development of the lymphatic system is thought to be a reaction to the inflammation and edema in the mucosa, aiming to dampen tissue damage by draining immune cells and excess fluid. However, similarly to blood vessels, lymph vessels show abnormal architectures and are often dysfunctional [209,210]. In CD, granulomatous structures characteristic of chronic lymphangitis have been observed in mesenteric lymph nodes and lymphatic vessels of the intestinal mucosa, and these correlate with disease activity [57,209,211]. In addition, lymphangiectasia, lymphadenopathy, and lymphatic vessel obstruction occurring during IBD compromise lymph drainage and leukocyte trafficking. This ultimately leads to edema, lymph leakage, and the deposition of adipose tissue into the mucosa, further fostering inflammation [211,212,213]. These observations are supported by the fact that a notable reduction in lymphatic contractile activity has been observed during murine experimental colitis, both locally and systematically [209,210,214]. Taken together, the lymphangiogenesis observed in IBD cannot resolve inflammation due the dysfunctionality of the newly formed lymphatics. The normalization of lymphatic function might represent an additional therapeutic approach in IBD.

### 3.7. Vascular Function and Extra-Intestinal Manifestations of IBD

More than one third of patients with CD and UC are affected by extraintestinal manifestations in addition to intestinal inflammation (Figure 2). The most common manifestations include thromboembolisms; hepatobiliary disorders; arthropathies; and cutaneous, pulmonary, and ocular manifestations, as well as neurological and psychosocial disturbances [14,215,216]. For example, major depressive disorder and multiple sclerosis are well-described IBD comorbidities [216,217,218]. Some of these extraintestinal manifestations correlate with flares of intestinal inflammation but others occur independently [215]. It remains unclear why only a fraction of patients with IBD present extra-intestinal manifestations.

#### 3.7.1. Systemic Vascular Barrier Dysfunction in IBD

The GVB has been shown to lie at the heart of the gut–liver–brain axis. Increased GVB leakiness and dysfunction has been observed in patients during *Salmonella* infection, diet-induced nonalcoholic steatohepatitis, and metastatic colorectal cancer, leading to an impaired gut–liver axis connection [13,219,220]. The breakdown of the GVB in IBD may induce widespread low-grade vascular inflammation through the uncontrolled release of microbial products or pro-inflammatory factors into the systemic circulation, which might then compromise the vascular barrier at distant organs and result in extra-intestinal manifestations [12]. A strong link between the GVB and the blood–brain barrier (BBB) has been established in IBD. Manifestations of anxiety and depression have been reported in up to 40% of patients with active IBD (14), together with deterioration in cognitive functions [216,217,218,221,222,223,224]. Similar observations have been made in DSS-colitis models where mice displayed increased anxiety- and depression-like behaviors and alterations of the limbic system [158,225,226,227]. Carloni et al. have recently shown that after DSS challenge, there is a persistent increase in intestinal vascular permeability in treated mice, even after recovery [158]. They also observed an increased absolute number and percentage of innate immune cells in the liver and the brain, suggesting that acute intestinal inflammation quickly spreads to other organs, including the brain [158]. Interestingly, vascular permeability induced by DSS treatment was only increased transiently in the brain due to the closure of the vascular barrier in the brain choroid plexus (PVB), which was then released after discontinuation of the application of DSS. In a transgenic mouse model of the inducible closure of the PVB, animals exhibited anxiety-like behavior and a deficit in short-term memory, suggesting that PVB closure may correlate with cognitive and mental disturbances [158]. These observations are further supported by the fact that a disruption of the BBB has been observed during TNBS-induced colitis in mice [228].

The translocation of bacteria into the bloodstream during IBD-related intestinal inflammation has been proposed to contribute to extra-intestinal effects. Indeed, patients with IBD have an elevated risk of sepsis [229,230]. Nevertheless, sepsis represents a rare complication of IBD, mainly occurring after surgery [230]. In DSS-induced colitis, mice treated with the glucocorticoid budesonide, bacterial translocation to the liver, and endotoxemia have been observed following massive intestinal barrier disruption [231]. However, these effects were not observed in the transfer colitis model, or in DSS-treated mice which did not receive budesonide. Together, these data suggest that the translocation of bacteria and bacterial products is limited during IBD and experimental colitis [231].

Bacterial products, on the contrary, have been frequently detected in the circulation of patients with IBD. Bacterial DNA can be found in the blood for up to 50% of IBD patients [232], and bacterial DNA translocation represents an independent risk factor of relapse at 6 months in CD patients [233]. The presence of bacterial endotoxin and LPS-binding protein can also be detected in the circulation of IBD patients [234,235,236,237,238]. Higher serum levels of LPS were observed in patients with IBD-associated spondylarthritis compared to IBD alone, indicating that an increased translocation of bacterial products is linked to the development of extra-intestinal manifestations [236]. Recently, Carloni et al. showed that the LPS blood concentration increases only transiently after DSS challenge in mice. The fact that patients with UC have elevated serum concentrations of LPS-binding protein but not of LPS supports the idea of a transient increase [158]. The authors proposed that increased scavenging by circulating inflammatory cells might explain the transient character of the serum LPS spike.

The release of inflammatory mediators by the inflamed intestinal endothelium might also lead to systemic low-grade inflammation, resulting in vascular activation and/or barrier dysfunction at distant organs. Increased levels of circulating inflammatory cytokines (IL-6, TNF-α…), for instance, can be found in IBD patients and animals with colitis [158]. There is indeed increasing evidence that circulating pro-inflammatory mediators in IBD patients may contribute to the progression of several central nervous system (CNS) disorders [158,239,240,241].

#### 3.7.2. Endothelial Damage and Systemic Vascular Inflammation

During IBD, local endothelial dysfunction, which can spread into systemic vascular barrier defects, might also promote generalized vascular inflammation. There is evidence linking IBD with atherosclerosis, coronary dysfunction, and an increased risk of cardiovascular (CV) morbidity and mortality [242,243,244,245,246,247,248]. Furthermore, IBD patients have an elevated risk of vasculitis, which is linked to more frequent headaches and extraintestinal symptoms [249,250]. Antibodies to endothelial cells reflect vascular injury and have been detected in the serum of patients with vasculitis. Anti-endothelial cell antibody levels are elevated in UC and CD patients compared with healthy controls [251]. In addition, increased anti-EC antibody levels were found to correlate with circulating levels of Von Willebrand factor, a marker of vascular inflammation and injury, indicating the occurrence of vasculitis during IBD. Some association between anti-endothelial cell antibody levels and disease activity has been found in both UC and CD [252,253].

Increasing evidence links intestinal microbiota dysbiosis, vascular aging, and cardiovascular diseases (the “gut–heart axis”). The passage of bacterial products into the bloodstream has been associated with increased arterial stiffness, atherosclerosis, hypertension, and cardiovascular risk in human individuals [254]. For instance, the reduction of short chain fatty acids and the increased production of trimethylamine-N-oxide, which are associated with gut dysbiosis, increase cardiovascular risk. In addition, the transmigration of LPS into the bloodstream activates vascular inflammation [254].

#### 3.7.3. Coagulation and Thrombosis in IBD

The local increase of pro-coagulant and pro-thrombotic events in the microvasculature of inflamed intestinal tissues is associated with systemic subclinical thrombosis in patients with IBD. Markers of coagulation are elevated in the serum of IBD patients, and increased extra-intestinal thrombus formation is enhanced in the DSS-colitis model [184,255]. Thrombosis represents a significant comorbidity, and thromboembolitic events have been estimated to account for up to 25% of IBD-related deaths [256]. The risk of venous thromboembolism is particularly increased in patients with IBD during a flare up or in the presence of chronically active inflammation [257,258,259,260,261,262,263]. In comparison, the overall risk of arterial disease is only modestly increased [260].

### 3.8. Targeting the Vasculature in IBD Therapy

The role of the microvasculature in the pathogenesis and perpetuation of IBD is receiving increasing recognition, and it represents an attractive therapy target.

Interestingly, several anti-inflammatory drugs used in the clinical management of IBD also ameliorate vascular dysfunction, suggesting that their efficacy depends in part on their endothelium-directed effects. For instance, mesalazine (5-aminosylicyclic acid), a medication used to treat mildly to moderately severe forms of IBD, has been shown to inhibit platelet activation [264]. In addition, the anti-TNF-α neutralizing antibody infliximab, which blocks TNF-induced inflammation and has been successfully used in IBD therapy, notably improves endothelial dysfunction in CD by enhancing agonist-induced vasodilatation, by reducing thrombus formation through inhibition of the CD40/CD40L/sCD40L pathway, and by inhibiting TNF-α-induced endothelial cell permeability [265,266,267].

Anti-TNF-α therapy has allowed healthcare professionals to bridge a therapeutic gap for IBD patients who are refractory or intolerant to treatment with classic immunosuppressive agents. However, a significant proportion of patients does not respond to anti-TNF-α therapy. New approaches based on the blockade of T-cell homing have shown promising results. Here, the recruitment of T-cells through the binding of α4β7 integrins to endothelial MadCAM1 is inhibited [268,269,270]. The α4β7-integrin-specific antibody vedolizumab induces long-term remission in CD and UC, and represents a good alternative for patients with refractory disease and colonic inflammation [271,272,273]. The β7-integrin-specific antibody etrolizumab as well as anti-MadCAM1 antibodies are currently being evaluated in clinical trials [269,274,275]. Hence, the specific blockade of the interaction of T-cells with the activated MadCAM1^+^ endothelium is increasingly being implemented in the clinical routine.

Therapeutic strategies targeting the endothelium represent an interesting way to reduce mucosal inflammation and/or extraintestinal manifestations by normalizing the vascular function. For example, heparin has been administrated to IBD patients to prevent venous thromboembolism [276,277,278]. In addition, treatment with low-molecular-weight heparin has shown therapeutic efficiency in IBD [279,280], which was not only due to the inhibition of microvascular thromboses, but also to immuno-modulating properties (such as the suppression of neutrophil recruitment) and to an increase in mucosal recovery [281,282]. A new therapeutic approach based on the delivery of heparin via nanoparticles (NPs) has provided promising results in the TNBS-induced colitis mouse model [283]. In addition, heparin-coated human serum albumin NPs have been shown to efficiently deliver drugs into the inflamed intestine in a murine model of colitis, opening up new possibilities for combinational treatment [284]. Similarly, the treatment of microvascular lesions with *Panax notoginseng* attenuated inflammation and disease activity in rats with colitis [285]. In another study, the targeting of phosphatidylserine externalized by stressed ECs in capillaries of the mouse colonic mucosa using annexin-V inhibited TNBS-induced colitis [286].

Another promising strategy for vessel-directed therapy of IBD is the inhibition of the loss of GVB function during IBD. The receptor tyrosine kinase imatinib has been shown to inhibit vascular dysfunction and edema in various models [162,287,288,289]. Treatment with imatinib blocks vessel permeability and alleviates DSS-induced colitis in mice [34]. Imatinib, given in the context of chronic myeloid leukemia, has been reported to induce long-standing remission of CD [290]. Sphingosine 1-phosphate (S1P), a sphingolipid mediator, represents another potential target. S1P signals through high-affinity G protein-coupled receptors S1P1 to 5 to regulate the egress of lymphocytes from lymphoid organs and the maintenance of vascular integrity [291,292], targeting both blood and lymphatic ECs. A dysfunctional S1P signaling axis leads to pathological angiogenesis and increased vascular permeability [293,294]. Several S1P agonists including ozanimod and etrasimod have shown promising results by blocking lymphocyte recruitment and improving barrier function, and are currently being tested in phase 3 clinical trials for UC and CD [293,295,296]. In conclusion, pharmacological normalization of the vasculature could not only prevent vascular co-morbidities in IBD patients, but might also complement the standard anti-inflammatory regimens.

## 4. Conclusions

The endothelium lies at the heart of the inflammation circle, and the manifold changes it undergoes during activation are regulated by complex mechanisms. These mechanisms can explain in part the refractory character of IBD, but might also represent complementary anti-inflammatory therapy targets. The blockade of leukocyte recruitment by endothelial cells plays an important role in that regard. Vascular damage, hyperpermeability, and the activation of the hyperthrombic state of the inflamed vasculature also represent important contributors to inflammation, whereas the role of angiogenesis appears to be less important than initially thought. In recent years, the intestinal microvascular barrier has been shown to play a decisive role as a second barrier in the gut. The loss of its function has furthermore been found to promote inflammation and has been linked to the development of extra-intestinal manifestations of IBD. Therefore, the intestinal microvascular barrier is now emerging as a promising therapeutic target for the treatment of IBD patients.

## Figures and Tables

**Figure 1 ijms-24-05517-f001:**
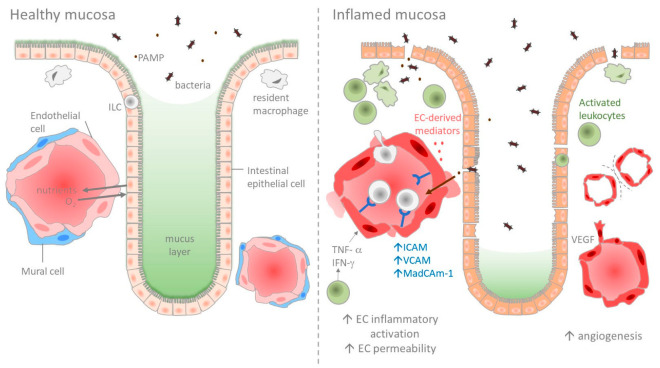
In the healthy intestine (depicted here, the colon mucosa), the mucosal microvasculature participates in homeostasis by regulating the absorption of essential nutrients, electrolytes, and water, while building a second barrier towards luminal microbes. During IBD, inflammation activates the intestinal microvasculature through release of cytokines and growth factors, leading to adhesion molecule expression, leukocyte extravasation, vascular hyperpermeability, and an increase in both sprouting and intussusceptive angiogenesis.

**Figure 2 ijms-24-05517-f002:**
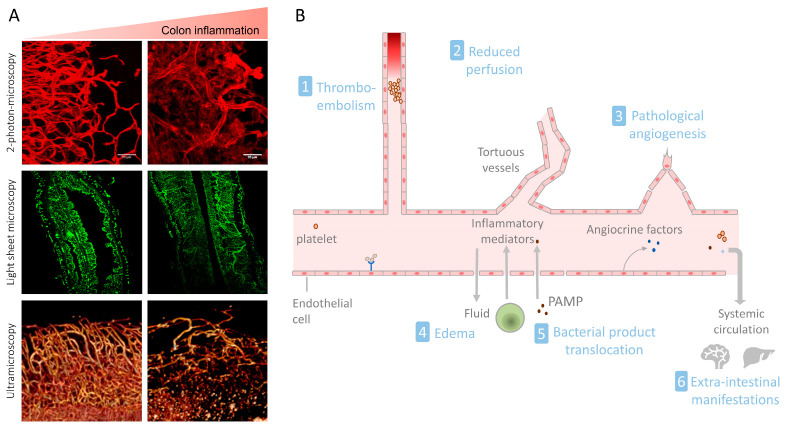
(**A**) The colonic microvascular architecture is massively remodeled in the presence of inflammation. The vessel structure was analyzed in mouse colon samples in the absence (left) or presence (right) of colitis. Vessels were visualized via the staining of CD31 (2-photon microscopy) or via lectin-staining (light sheet microscopy and ultramicroscopy). (**B**). Vascular changes and dysfunction observed in IBD participate in the initiation and perpetuation of mucosal inflammation and extra-cellular manifestations. PAMP: pathogen-associated molecular pattern.

**Table 1 ijms-24-05517-t001:** Different mesenteric vascular cell subtypes in homeostasis and in IBD.

Cell Type	In Homeostasis	In IBD
HIMECs	Tolerance to bacterial products from the gut microbiota Constitutive iNOS expression	Leukocyte hyperadhesion Angiogenesis ↑ ανβ3 integrin expression ↑ vessel permeability ↓ iNOS and eNOS expression ↓ protein C system activation ↑ secretion of inflammatory mediators
HEVs	Recruitment and trafficking of lymphocytes from blood to lymph nodes and secondary lymphoid organs	↑ density ↑ leukocyte binding Formation of extrafollicular HEVs
Lymphatic ECs	Absorption of fatty acids Immune regulation	↑ density ↓ contractile activity Lymphangitis
Mural cells	Development and maintenance of the GVB	↓ vessel coverage ↑ MMP expression
Enteric glial cells	Part of gut–vascular units Development and maintenance of the GVB	Sensing of bacterial translocation Closure of the PVB

Abbreviations: ECs: endothelial cells; HIMECs: human intestinal microvascular ECs; HEVs: high endothelial venules; TJs: tight junctions; GVB: gut–vascular barrier; PVB: plexus–vascular barrier; ↑: increase; ↓: decrease.

**Table 2 ijms-24-05517-t002:** Vessel-directed effects of key inflammatory cytokines involved in the pathogenesis of IBD.

Cytokines	Effect
IFN-γ	EC activation, ↑ CAM expression (notably MadCAM-1) ↑ vascular permeability Disassembly of VE-cadherin junctions, ↓ VE-cadherin expression ↓ EC proliferation and migration, ↓ angiogenesis ↓ vascular coverage, ↓ PDGF-B ↑ TLR3 expression ↑ CX3CL1 (fractalkine)
TNF-α	EC activation, ↑ CAM expression (notably MadCAM-1) ↑ vascular permeability, ↑ Phosphorylation of VE-cadherin, ↑ monolayer tension ↓ TJ protein expression in EC ↑ circulating levels, ↑ vascular dysfunction ↓ EC proliferation and migration, ↓ angiogenesis ↑ CX3CL1 (fractalkine)
IL-1β	EC activation, ↑ CAM expression (notably MadCAM-1) ↑ CX3CL1 (fractalkine)

Abbreviations: ↑: increase; ↓: decrease.

**Table 3 ijms-24-05517-t003:** Pathogenic effects of vascular factors in IBD.

Factor	Effect on the Intestinal Vasculature
Angiogenic growth factors	
*VEGF*	Expression increased in IBD ↑ sprouting angiogenesis ↑ EC proliferation and migration ↑ ICAM-1 and VCAM-1 expression ↑ recruitment of VEGFR- expressing immune cells ↑ vascular permeability, disassembly of VE-cadherin junctions ↓ vascular coverage ↑ wound healing
*bFGF*	Expression increased in IBD ↑ sprouting angiogenesis
*PDGF*	Expression increased in IBD ↑ sprouting angiogenesis ↑ vascular coverage Protective effect in UC
Nitric oxide (NO)	Decreased constitutive expression of eNOS and iNOS in intestinal EC during IBD ↓ CAM expression ↓ ROS production ↑ intestinal endothelial barrier function ↑ vasodilatation Potentiates VEGF-mediated effects
Coagulation factors	Increased platelet activation and thrombi in IBD ↓ thrombomodulin expression in IBD ↓ protein C receptor expression in IBD Impaired protein C activation in activated intestinal EC
Toll-like receptors (TLR)	Tolerance to endotoxin in intestinal EC Expression of TLR3 and TLR5 by intestinal EC, protective against colitis in mice ↑ endothelial barrier function
Angiocrine factors	↑ CX3CL1 (fractalkine) → ↑ adhesion and activation of CX3CR1+ leukocytes ↑ CL25 → recruitment of CCR9+ immune cells → protective effect ↑ NO ↑ CXCL10 → epithelial cell survival

Abbreviations: ↑: increase; ↓: decrease.

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
