# Peer review of "Contribution of Blood Vessel Activation, Remodeling and Barrier Function to Inflammatory Bowel Diseases"

_ijms, 2023, doi:10.3390/ijms24065517_

Round 1

Reviewer 1 Report

This article discusses the involvement of blood vascular activation, remodelling, and barrier function in the pathophysiology of inflammatory bowel disease. The review's importance to the readership is clearly and successfully highlighted. The authors rightly note that their topic has not gotten enough attention in recent literature. The review's aims are expressed in a general yet appropriate manner. References are employed effectively to support important claims. I applaud the authors for their many strengths, including tackling a fascinating and important subject.

The paper is already well written and has clinical value in addition to being within the scope of the journal. Notwithstanding these advantages, a few points in the article are worth further discussing.

Clarifying these, in my opinion, will strengthen the article even more:

·         It is unclear where the results of human clinical research end and those of animal experiments begin in the material supplied by the authors. Chemical models of experimental colitis are more effective in studying the effects of therapeutic drugs than complex pathomechanism. It is essential to be cautious when interpreting the results of relatively basic animal experimental models and not to equate not equate them with the actual pathophysiological pathways of the disease in humans, and I believe the authors do so. The authors should make a clear distinction between data from experimental research and data from human studies. They might do this by arranging them in distinct sub-chapters.

·         My other, maybe more subjective, impression is that the authors underestimate the relevance of the microbiota and its alterations in the pathogenesis of IBD, in terms of the topics they discuss, of course.

Minor

The authors do not explain the abbreviation EC . I understand that they mean endothelial cells but this abbreviation can have different meanings also in the context of the intestine where often such an abbreviation is used to refer to epithelial cells.

Author Response

We thank the reviewer for their time and effort, as well as their valuable suggestions.

You can find below our point-by-point response to the reviewer's comments:

1) We revised our manuscript and systematically stated whether the studies reported were obtained from IBD patients or animal models. This information is highlighted in pink in the revised manuscript. For clarity, we grouped human or animal data together (see points 3.3.1, 3.5.2, 3.6 and 3.7.1). 

2) According to the reviewer's suggestion, the relevance of the intestinal microbiota for the vasculature in IBD has been discussed in the revised manuscript (see points 1, 3.3, 3.5, 3.7.1 and 3.7.2).

3) We have defined the abbreviation ECs (endothelial cells) in the revised manuscript (page 2, line 48).

Reviewer 2 Report

I commend the Authors for their nice and comprehensive work. I have just few suggestions:

- I would include a figure or a table summarising all the intestinal EC subtypes the Authors mentioned in their work. Authors may also consider to include the possible role of each EC subtype in IBD.

- I would stress the importance of intestinal microbiota as a regulatory factor of vascular function and as a bridge between inflammation and EC activation (PMID 28743984). A recent systematic review has investigated the association between gut microbiota and arterial stiffness, a well known marker of vascular ageing and vascular dysfunction, reporting associations between arterial stiffness and certain characteristics of intestinal microbiota such as variety and abundancy (PMID 35743626). Future studies systematically assessing vascular function and microbiota composition are warranted to identify potential novel prognostic and therapeutic markers.

Author Response

We are thankful to the reviewer for his/her time and valuable suggestions.

Changes in our manuscript has been highlighted in yellow. You can find below our point-by-point response:

1) We added a table (Table 1) summarizing all the intestinal EC subtypes mentioned in our manuscript, and their role under homeostasis or during IBD.

2) Following the reviewer's suggestion, we highlighted the role of the intestinal microbiota for the vasculature in IBD in the revised manuscript (see points 1, 3.3, 3.5, 3.7.1 and 3.7.2), and cited the suggested references.

Reviewer 3 Report

This review article describes ‘Contribution of blood vessel activation, remodeling and barrier function to inflammatory bowel diseases’.  You summarized well about the review of IBD as aspects of both clinical and basical sciences in it.  However, I point out the issues to be addressed.

Major

i)               Page 7, on 3.3 the gut vascular barrier in IBD, and/or Page 7, on 3.3.1 VE-cadherin and vascular barrier regulation in IBD, it seems that you should add the information of the intestinal permeability due to make clear the function of inflammation.  As you know, finally, the microvascular affects to the intestinal permeability and the intestinal permeability affects to recovery to the intestinal tissues on diseases condition such as UC and CD.  And also the intestinal permeability must have the barrier function from the intestinal bacteria and so on. I suggest following paper under below to cite your review article. 

Comparison of the intestinal drug permeation and accumulation between normal human intestinal tissues and human intestinal tissues with ulcerative colitis.

Nakai D, Miyake M, Hashimoto A.J Pharm Sci. 2020,109(4):1623-1626.

The change of the electrophysiological parameters using human intestinal tissues from ulcerative colitis and Crohn's disease.

Nakai D, Miyake M.J Pharmacol Sci. 2022,150(2):90-93.

Minor

i)               Page 13, Line from 558 to 559 , it seems you made a mistake on paragraph.

Author Response

We are thankful to the reviewer for his/her time and valuable suggestions.

Changes in our manuscript has been highlighted in yellow. You can find below our point-by-point response:

i) According to the reviewer's suggestion, we have highlighted the importance of the epithelial barrier disruption in IBD pathogenesis (see points 1 and 3.3 and references 116-118).